# Determination of the Strongest Factor and Component in a Relationship between Lower-Extremity Assessment Protocol and Patient-Oriented Outcomes in Individuals with Anterior Cruciate Ligament Reconstruction: A Pilot Study

**DOI:** 10.3390/ijerph18158053

**Published:** 2021-07-29

**Authors:** Hyung Gyu Jeon, Byong Hun Kim, Tae Kyu Kang, Hee Seong Jeong, Sae Yong Lee

**Affiliations:** 1Department of Physical Education, Yonsei University, Seoul 03722, Korea; hgjeon@yonsei.ac.kr (H.G.J.); bh_kim@yonsei.ac.kr (B.H.K.); tk82@yonsei.ac.kr (T.K.K.); 2International Olympic Committee Research Centre Korea, Seoul 03722, Korea; 3Institute of Convergence Science, Yonsei University, Seoul 03722, Korea

**Keywords:** knee injury, return to play, physical functional performance, muscle strength, neuromuscular control, muscle fatigue

## Abstract

Although the Lower-Extremity Assessment Protocol (LEAP) assesses multidimensional aspects of a patient with anterior cruciate ligament (ACL) injury, there is a need to reduce the dimensionality of LEAP items to effectively assess patients. Therefore, the present study aimed to establish the validity of LEAP and to determine associated factors and components in a relationship between LEAP and the International Knee Documentation Committee (IKDC) questionnaire. Fifteen patients who had ACL reconstruction more than 1 year and less than 5 years earlier were recruited. Patients performed LEAP, including muscular strength, fatigue index, static balance, drop landing, and functional hopping assessment. They also completed the IKDC questionnaire and the Tegner Activity Score. Factor analysis and stepwise regression analysis were performed. The 14 components of LEAP were categorized into four factors (functional task, muscle strength, neuromuscular control, and fatigue), which accounted for 83.8% of the cumulative variance by factor analysis. In the stepwise regression analysis, the functional task (*R*^2^ = 0.43, *p* = 0.008) in factors and single-leg hop (*R*^2^ = 0.49, *p* = 0.004) in components were associated with patient-oriented outcomes, respectively. In conclusion, the functional task and single-leg hop may be used for providing valuable information about knee joints to patients and clinicians.

## 1. Introduction

Anterior cruciate ligament (ACL) injury is a common sports-related musculoskeletal issue, not only in professional/elite athletes but also in recreational players [1,2]. ACL injury usually results from non-contact mechanisms (75.9%) observed in sports such as basketball, soccer, football, gymnastics, and field hockey [3,4,5,6]. Mechanisms include a rapid change of direction, jumping, pivoting, and cutting tasks [3]. ACL rupture causes subsequent functional defects, including knee joint instability and loss of function during dynamic tasks [7]. Furthermore, altered osteo- and/or arthro-kinematics, which are tibial anterior translation and knee internal rotation, are usually observed in those with ACL injury and result in knee joint instability [8,9].

Based on these factors, ACL reconstruction (ACLR) surgery is performed on patients who want to restore the altered movement patterns and function of their knee joints, increase the possibility of return to play (RTP), and reduce the possibility of post-traumatic osteoarthritis [10]. As a result, patients who want to participate in activities of daily living and high-intensity sports require ACLR surgery and careful rehabilitation to ensure proper recovery of the knee joint [11,12,13,14,15,16,17,18]. Accordingly, the goal of ACLR surgery is to maximize the stability and functional ability in the ACL-deficient knee and to promote a return to pre-injury activity levels and sports participation [19].

There are several deficiencies that can occur after ACLR when compared with the patient’s condition before the injury. For instance, common symptoms after ACLR include atrophy, strength deficit of quadriceps and hamstring, and decreased neuromuscular control [20,21,22,23,24]. Further evidence for a deficiency and lower rate of RTP, despite successful surgery, is that only half of the participants returned to competitive sports [12]. More importantly, patients with a history of ACL injury are more likely to suffer a second ACL injury within a few years [25]. Because of these factors, a step-by-step rehabilitation program should be conducted, and more importantly, assessment is needed to evaluate whether a deficit have been eliminated from a variety of perspectives before RTP.

Recently, Difabio et al. [26] have used the Lower-Extremity Assessment Protocol (LEAP), which uses muscle strength and functional tests, single-leg balance, and the Landing Error Scoring System (LESS) to assess the function and deficits of the knee joint and lower extremity after ACLR. This study concluded that LEAP could provide unique information through a series of functional tests used to assess lower-extremity function after ACLR. LEAP has been used to provide comprehensive information for patients and clinicians; however, it is not clear whether this protocol includes redundant tests. In addition, duplicated components of LEAP may require excessive time and cost for patients and clinicians. Therefore, we aimed to determine which elements of LEAP could be the most powerful tools to detect patients’ deficits and symptoms that could indicate a potential risk of second ACL injury.

Subjective self-reported pain and function are the indicators that best represent the patient’s experience (e.g., International Knee Documentation Committee [IKDC] questionnaire); thus, patient-oriented outcomes are investigated through questionnaires and used as the evidence and/or criteria for determining RTP for patients with ACL injuries [27,28]. For assessing readiness for RTP in a patient after ACLR, LEAP assesses multidimensional aspects of function, muscle strength, balance, and fatigue. It is necessary to examine which measurable evaluation factors reflect the patient’s self-reported results. In addition, the validity of the LEAP test should be addressed to verify whether the battery of test represents each component of risk factors. Therefore, the purposes of this study were (1) to establish the validity of the components of LEAP, (2) to determine the factor of LEAP that has the strongest relationship with patients’ function and pain, and (3) to simplify LEAP by determining the associated component within the strongest association factor.

## 2. Method

This study was a descriptive and cross-sectional design in a laboratory setting.

### 2.1. Participants

The participants in this investigation were also studied by Kim et al. [29]; however, this analysis was independently conducted. A total of 15 individuals with a history of ACLR participated in this study (Table 1). The inclusion criteria used to recruit participants were as follows:Aged 19 to 40 years;More than 1 year and less than 5 years after ACLR (patients after ACLR surgery have a greater risk of suffering a subsequent ACL injury and the possibility of post-traumatic osteoarthritis [25,30]; thus, we limited the elapsed time after ACLR surgery to focus on the ACL injury);Verified history of ACLR surgery; andACLR surgery with autograft.

Participants excluded from the study had the following:A history of lower-extremity injuries within the past 3 months;A history of surgery in the lower extremities except for ACLR;A history of ACL re-injury (only patients who underwent ACLR due to an initial injury were included); andDamage or impairment affecting the muscle and nerve function.

All participants were informed of the study method and procedure and signed an informed consent form approved by the Institutional Review Board of Yonsei University (7001988-201804-HR-356-02).

### 2.2. Procedure

Each participant performed LEAP, including functional hopping task, muscle strength, fatigue index, LESS, and static single-leg balance. Participants completed the IKDC questionnaire to assess the functional status and pain level of the knee joint during activities of daily life and sports activities. In addition, the Tegner Activity Score was used to determine whether the participant had already successfully returned to their daily activities or sports activities and to measure their current physical activity level.

### 2.3. Strength Assessment and Fatigue Index

After warming up for 10 min, the isokinetic and isometric strength of the knee joint were measured using an isokinetic dynamometer (ConTrex AG, Dubendorf, Switzerland). Participants were tested in a sitting position, according to the manual of the dynamometer manufacturer (Figure 1A). The upper body was fixed by a seatbelt, and the unmeasured leg was fixed by a bar for stabilization. Each participant was instructed to concentrically contract for knee flexion (hamstrings) and extension (quadriceps), and torque was measured at angular velocities of 60°/s and 180°/s through a preset range of knee motion within safety limits specific to each participant. After three practice movements, the peak torque per body weight was recorded.

Participants were instructed to perform maximal voluntary isometric contractions while fixing their chest on the dynamometer and holding hands to measure isometric muscle strength and fatigue index of quadriceps and hamstrings. While participants were performing isometric muscle contractions of knee flexion and extension, the peak torque value was recorded to identify isometric muscle strength. All values of muscle strength were normalized with respect to each participants’ body weight. An isometric contraction for 30 s was measured, and muscle strength for the first 5 s was compared with that for the last 5 s to identify how long the muscle could maintain its strength (muscle fatigue index). Participants were given advance verbal instructions and encouragement to push or pull as hard as possible to facilitate maximal effort during muscle strength assessment.

### 2.4. Functional Assessment

The functional test consisted of four types of hopping tasks: (1) single hop for distance, (2) triple hop for distance, (3) cross-over hop for distance, and (4) 6-m timed hop [31]. All components of the functional test were performed three times at the indicated start line (Figure 1B). During the functional test, participants fixed their hands on their hips without swinging their arms. Distance was remeasured when the lifted leg touched the floor. The first three hop tests were recorded as the distance (m) from the starting line, and the 6 m timed hop was recorded as the time of task completion (s).

### 2.5. Balance Assessment

The single-leg standing assessment was carried out to determine the center of pressure velocity on the force plate (AccuSway plus balance force plate, AMTI, Watertown, MA, USA) using analysis software (Balance Clinic version 1.4.4, AMTI, Watertown, MA, USA). Participants stood on a single leg in the center of the plate for 10 s with their eyes closed and their hands on their hips (Figure 1C). After some practice, the balance assessment was performed. The assessment was remeasured when the participant opened their eyes, or the other leg touched the floor, or the individual completely lost their balance or jumped or dragged their foot to restore their balance. Data were recorded at 50 Hz.

### 2.6. Jump Landing Assessment

The LESS is a cost-effective clinical assessment tool to determine the potential risks of a jump landing task [32]. Two video cameras (front and side views) were used. Participants were instructed to jump to a half-height from a 30 cm box and land at the landing point and then to jump again with their maximum vertical jump (Figure 1D). Before the evaluation, participants had three opportunities to practice the jump landing task. The evaluator only explained the procedure and did not coach any landing technique to record the participant’s natural movement pattern and quality. For three successive trials, the front camera and the side camera were located 3.5 m from the landing point and 1.2 m high to capture the lower-extremity motion during the jump landing motion. LESS scoring meant the number of errors during the jump landing movement could be easily observed, and the maximum error score was 19 points.

Assessment items included lower extremity and trunk positioning at the time of initial contact: (1) maximum knee flexion (less than 30° = error score 1); (2) hip flexion (thigh was in line with the trunk = error score 1); (3) trunk flexion (trunk was vertical or extended on the hips = error score 1); (4) ankle plantar flexion (foot lands heel to toe or with a flat foot = error score 1); (5) medial knee position (center of the patella point towards the medial of midfoot = error score 1); (6) lateral trunk flexion (midline of the trunk is flexed to the left or the right = error score 1); (7) stance width, i.e., wide or narrow, (feet are positioned greater or less than acromion process = error score 1); (8) toe in or out (foot position is internally or externally rotated more than 30° from initial contact to maximum knee flexion = error score 1); (9) symmetric initial foot contact (both feet did not land at the same time or landing pattern of each foot was not the same = error score 1) [32]. Furthermore, movement displacement was also assessed: (1) knee flexion displacement (knee flexes less than 45° between initial contact and maximum knee flexion = error score 1); (2) hip flexion displacement (thigh did not flex more on the trunk from the point of initial contact to maximum knee flexion = error score 1); (3) trunk flexion displacement (trunk did not flex more from the point of initial contact to maximum knee flexion = error score 1); (4) medial knee displacement (center of the patella point towards the medial of midfoot at maximum knee valgus position = error score 1) [32]. Lastly, overall trunk and lower-extremity displacement (using a large amount of trunk, hip, and knee displacement = error score 0, soft landing; using some amount of, trunk, hip, and knee displacement = error score 1, average; using very little trunk, hip, and knee displacement = error score 2, stiff landing) and impression (soft landing with no frontal-plane or transverse plane motion = error score 0, excellent; all landing pattern that not applicable to excellent and poor = error score 1, average; using large frontal-plane or transverse-plane motion or stiff landing with some frontal-plane or transverse-plane motion = error score 2, poor) was assessed [32].

The lower-extremity landing strategies for the hips, knees, and trunk that could be a risk factor for the knee joint were evaluated through various aspects in the front and side views. The evaluator used motion-analysis Kinovea software (version 0.8.15) for scoring.

### 2.7. Statistical Analysis

Exploratory factor analysis was performed for the 14 components of LEAP, namely, the single hop, the triple hop, the cross-over hop, the 6 m timed hop, isokinetic flexion 180°/s, isokinetic extension 180°/s, isokinetic flexion 90°/s, isokinetic extension 90°/s, isometric contraction of quadriceps, isometric contraction of hamstring, single-leg balance, LESS, the hamstring fatigue index, and the quadriceps fatigue index. LEAP was validated by grouping similar variables into dimensions. Pearson’s correlation analysis was conducted using the 14 components of LEAP. Within each of the four factors, the corresponding raw data were added, and stepwise regression analysis was used to determine the factor that most affected the IKDC. Then, stepwise regression analysis was once again conducted to detect the most important components to simplify the assessment procedure within the strongest factor. All statistical analyses were conducted using SPSS statistics program version 25.0 (IBM Corp., Armonk, New York, NY, USA). The alpha level for all analyses was set at 0.05.

## 3. Results

### 3.1. Categorizing Results of the Lower-Extremity Assessment Protocol

All independent variables of LEAP were analyzed using factor analysis because there were no items that inhibited validity. Therefore, the 14 components of LEAP were the factors analyzed using principal components analysis with Varimax rotation. The factor loadings were all above 0.5, satisfying the validity of the overall LEAP components. The analysis yielded four factors that explained 85.315% of the variance for the entire set of variables (Table 2). The first factor was labeled “functional task” with the following components: the single hop, triple hop, cross-over hop, and 6 m timed hop (explained 30.335 of the variance). The second factor included the most components of the four factors, and it was labeled “muscle strength” with the following components: isokinetic knee flexion strength (90°/s and 180°/s), isokinetic knee extension strength (90°/s and 180°/s), and isometric knee flexion and extension strength (explained 28.492 of the variance). The third derived factor was labeled “neuromuscular control” with the following components: single-leg balance and LESS (explained 14.884 of the variance). The fourth factor was labeled “fatigue index” and included knee flexion fatigue index and knee extension fatigue index (explained 11.604 of the variance). The results of Pearson’s correlation are illustrated as a matrix (Figure 2).

### 3.2. The Strongest Factor of the Lower-Extremity Assessment Protocol

Multiple stepwise regression analyses were used to test if the factors and components of LEAP significantly predicted the IKDC questionnaire score. Predictors included the functional hopping task, muscle strength, neuromuscular control, and fatigue index (Factors 1–4). However, only the functional hopping task factor was inputted, and the other predictors were excluded. A significant regression equation was found (*F*_(1,13)_ = 9.88, *p* = 0.008), with an *R*^2^ of 0.43 (Table 3). Predicted patient-oriented outcome was equal to 43.85 + 0.03 (sum of functional hop task). The patient-oriented outcome increased by 0.03 points for the sum of functional hopping task factors.

### 3.3. The Strongest Component of the Lower-Extremity Assessment Protocol

Multiple stepwise regression analyses of LEAP components included the single hop, triple hop, cross-over hop, and 6 m timed hop to significantly predict the IKDC questionnaire score. However, only the single-hop task component was inputted, and other functional task components were excluded. A significant regression equation was found (*F*_(1,13)_ = 12.57, *p* = 0.004), with an *R*^2^ of 0.49 (Table 4). Predicted patient-oriented outcome score was equal to 42.82 + 0.26 (single-hop task). The patient-oriented outcome increased by 0.26 points for each centimeter of the single-hop task. Respectively, the hopping task function and single-hop task were significant predictors of patient-oriented outcomes as a factor and component of LEAP.

## 4. Discussion

The purpose of this study was to establish the validity of LEAP by classifying the components of LEAP and identifying the strongest factor and component on patient-oriented outcomes, including pain and function. As a result, LEAP was divided into four factors (functional hopping task, muscle strength, neuromuscular control, and fatigue index). Functional hopping and the single-hop task have been reported as the strongest factor and component, respectively, on patient-oriented outcomes after ACLR.

The findings of the present study were consistent with those mentioned in previous studies, in that the “functional task” was one of the strongest predictors of patient-oriented outcomes. Logerstedt et al. [33] reported that the single hop, cross-over hop, triple hop, and 6 m timed hop at 6 months after ACLR were significant predictors of self-reported knee function at 1 year after ACLR. Although all factors of LEAP were important and essential for RTP, the functional task factor of hopping was a comprehensive representation of the properties of all factors. There are several possible explanations for such a result. First, sufficient muscle strength of the lower extremity is essential to perform a forceful and energetic task with more distance and correct motion not related to the injury mechanism. In addition to adequate muscle strength, a complete range of motion is necessary to use a sufficient shock-absorbing strategy in strong jumping and landing movements. In more detail, when performing the functional task, it is necessary to include eccentric contraction to load the force in preparation for a strong jump. After the eccentric contraction, all the joints are extended, and the force is produced through the concentric contraction. Furthermore, patients will not be able to record a sufficient jump distance if they are unable to distribute shock through sufficient joint flexion or if the knee joint is incomplete at landing. Since the functional task is initiated using a single leg, including standing in the starting position, leaping, and landing, the functional task of hopping requires static and dynamic balancing capabilities to maintain the balance of the body. Finally, to evaluate a patient’s RTP decision, not only the strength of the knee joint but also the strength of the other lower extremity and core are needed, and all of them must be coordinated as an optimal kinetic chain system. However, the strength assessment component of LEAP includes only the knee joint muscle strength, not the other lower extremity. Therefore, functional tasks are more valuable for ACLR patients because if any deficiencies exist in other parts of the body, the patient may not be able to perform the functional task. For the above reasons, the “functional task” can reflect the overall function of the lower extremity, comprehensively includes other factors, and is directly proportional the patient’s function and pain for the RTP decision.

The results of this study demonstrate that the strongest component in the four functional tasks is the single-hop task. These findings correspond with the results of previous studies that have reported that the single-leg hop task correlated with results of isokinetic knee extension/flexion strength after ACLR [34,35]. Likewise, Logerstedt et al. [33] demonstrated that the single-leg hop task was one of the significant predictors of self-reported knee function. Additionally, previous studies [33,34,35] reported that a single-leg hop task seemed to be correlated with the self-reported questionnaire and muscle strength, but it differed from the current study in that it analyzed the patient’s pain and function in comparison to the various assessment dimensions of LEAP. In conclusion, the single-leg hop task could be used as a performance-based assessment to evaluate the combination of the ability to withstand loads related to sports-related activities, including muscle strength, neuromuscular control, and single-leg balance.

After ACL injury or ACLR, all lower-extremity muscles should be included in the assessment, but the quadriceps muscle should be highlighted because it has a direct connection tissue with the knee joint through the patella tendon. Furthermore, quadriceps muscle weakness and atrophy are common symptoms after ACL injury or ACLR [20,21]. Consequently, several systematic reviews demonstrated that quadriceps and hamstring muscle strength assessment were used as objective criteria to evaluate recovery after ACLR [36,37]. The strength of the quadriceps muscle is also an important measurement of patient satisfaction and is associated with RTP after ACLR [37,38,39]. Assessment of the hamstring as well as quadriceps muscles is also necessary, because hamstring muscles play an important role in protecting the ACL when the ACL collapses by performing anterior translation limitation of the tibia relative to the femur as an ACL agonist [40]. Additionally, knee medial flexor muscles, including semitendinosus and semimembranosus, contribute to knee joint stability and inhibit excessive valgus movement [41]. Della Villa’s study [42] demonstrated the importance of quadriceps and hamstring strength, highlighting that the deficit in both muscles was related to lower IKDC score during recovery from ACLR. Therefore, training and evaluating the strength of the quadriceps and hamstring are essential procedures for patients after ACLR prior to determining RTP.

After ACLR, neuromuscular control during dynamic movement, such as a landing task, is altered [41,43,44]. A previous cohort study demonstrated that knee motion, such as increased knee abduction angle at initial contact and reduced maximum knee flexion angle during landing, was consistent with peak vertical ground reaction force, which increased the loading on the knee and exposed the patient to ACL injury [44]. The observed knee movement with these biomechanical characteristics may demonstrate a lack of neuromuscular control of the lower extremity. The landing movement pattern or quality, including foot, trunk, and whole-body position, which can be assessed through LESS, may affect ACL injury and/or re-injury and other lower-extremity injuries [32]. Additionally, individuals with abnormal or altered neuromuscular control may show limited ability in the dynamic task of supporting the center of mass over the base after ACLR [43], which could cause dynamic instability with recurrent episodes of joint subluxation [36]. As a result, altered neuromuscular control, including a postural deficit, is a predictor of ACL re-injury during sports activities following RTP [43]. Therefore, neuromuscular control should be assessed before RTP in sports that include landing, pivoting, and cutting maneuvers consistent with ACL injury mechanisms to prevent re-injury. If neuromuscular control deficits are identified through single-leg balancing or LESS, neuromuscular training may improve the control of abnormal joint translation in muscle activity patterns during functional activity.

The ability to adapt and overcome muscle fatigue is one of the factors to be considered before determining RTP. Previous studies have shown that muscular fatigue not only decreases the ability to detect joint motion but also exhausts the afferent and efferent pathways, interrupting the compensatory stabilizing mechanism of the knee joint and force production [45,46]. These altered characteristics could increase anterior, posterior, and anterior–posterior joint laxity during sports activities or exercise in both males and females [47,48,49]. Since muscle fatigue causes joint instability in the second half of the period when fatigue is accumulated during activity or sports participation, the ability to adapt to fatigue should be improved before RTP.

This study has some limitations that must be recognized. First, despite graft selection being one of the factors that affects recovery progress, especially for the strength of knee joint [50], graft type was not controlled in the present study. However, the characteristics of patients, including surgery duration, verification of ACLR history, and physical activity level, were tightly controlled to establish the validity of LEAP. Second, since this study had a small sample size, results are not highly reliable and cannot be generalized for a larger population. However, since conducting a pilot study in the field of sports medicine is valuable in terms of the effective use of limited resources and appropriate interpretation of results [51], our study could establish a basis for future research regarding rigorous assessment in individuals with ACLR for RTP. In future research, it is necessary to verify the categorized factors and components evaluated in this study by conducting confirmative factor analysis using additional sample sizes. Third, we applied LEAP to active individuals before identifying the most influential risk factor by applying LEAP to athletes. Therefore, further study is needed to apply LEAP to athletes and to investigate dynamic tasks that may provide additional valuable information about knee joints in relation to functional tasks for determining RTP after ACLR. Lastly, although several previous studies have scored LESS using Kinovea software [52,53], care must be taken while interpreting the results, as Kinovea is a 2D assessment tool. Nevertheless, we expect that this study may provide useful information to both clinicians and patients by examining elements of various dimensions of the knee joint.

## 5. Conclusions

LEAP for RTP is categorized into four factors, and the functional tasks of hopping and the single-leg hop task are the strongest factor and component, respectively, for function and pain in patients after ACLR. These results suggest that the functional task of hopping and the single-leg hop task may be used to provide valuable information to both patients and clinicians, because the function and pain of the patient should be considered first when the clinician determines RTP after ACLR. In addition, since the functional task is the strongest factor, it is necessary to review dynamic tasks for patients after ACLR in a future study.

## Figures and Tables

**Figure 1 ijerph-18-08053-f001:**
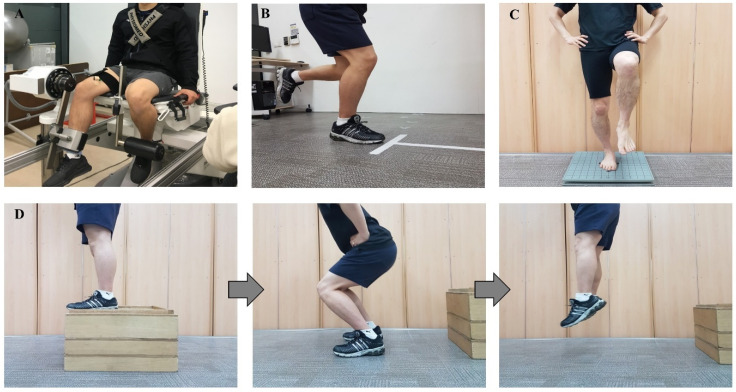
Procedure of Lower-Extremity Assessment Protocol. (**A**) isokinetic and isometric strength assessment of knee joint; (**B**) functional hopping task; (**C**) single-leg balance on force plate; (**D**) Landing Error Scoring System.

**Figure 2 ijerph-18-08053-f002:**
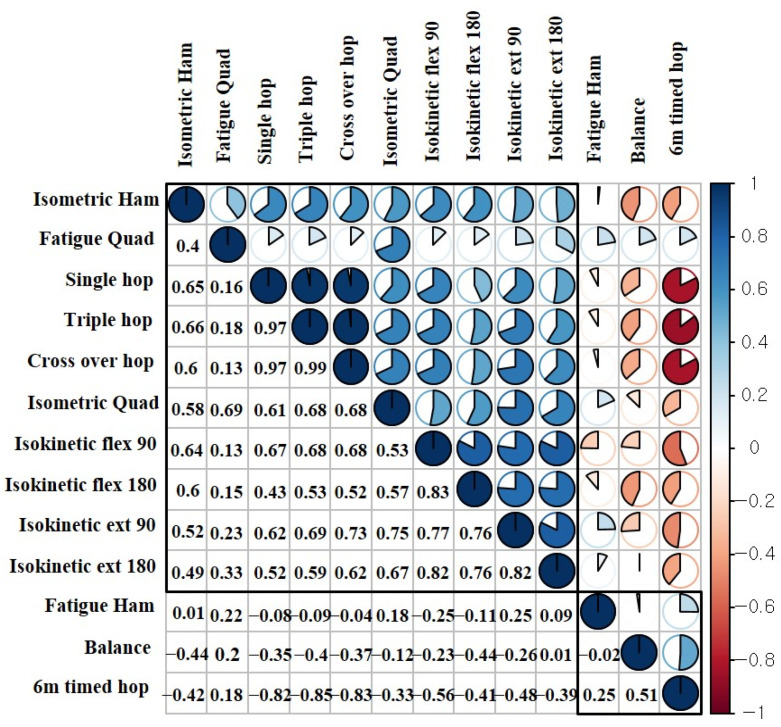
Pearson correlation matrix of the components of the Lower-Extremity Assessment Protocol. Abbreviation: Ext, extension; flex, flexion; ham, hamstring; quad, quadriceps.

**Table 1 ijerph-18-08053-t001:** Characteristics of patients after ACL reconstruction. Data are expressed as mean and standard deviation, except for sex (expressed as numbers).

	ACLR (*n* = 15)
Age (years)	27.87 ± 4.10
Sex (m:f)	12:3
Height (cm)	172.56 ± 4.81
Weight (kg)	75.51 ± 13.03
Time to surgery (months)	27.11 ± 14.03
IKDC score (percentage)	71.87 ± 16.60
Tegner activity level (current)	5.73 ± 1.16

The score of IKDC was transformed into percentages. Abbreviation: IKDC, International Knee Documentation Committee.

**Table 2 ijerph-18-08053-t002:** Summary of factor analysis for the Lower-Extremity Assessment Protocol.

	**Components**	**Factor Loading**			
		**Factor 1**	**Factor 2**	**Factor 3**	**Factor 4**
**Functional task**	Single hop	**0.926**	0.296	−0.131	0.007
	Triple hop	**0.900**	0.374	−0.184	0.031
	Cross-over hop	**0.877**	0.389	−0.188	0.035
	6 m timed hop	**−0.784**	−0.198	0.422	0.267
**Muscle strength**	Isokinetic knee flexion 180°/s	0.144	**0.917**	−0.265	−0.018
	Isokinetic knee extension 180°/s	0.310	**0.868**	0.130	0.087
	Isokinetic knee flexion 90°/s	0.407	**0.849**	−0.070	−0.212
	Isokinetic knee extension 90°/s	0.389	**0.778**	−0.155	0.262
	Isometric knee extension	0.547	**0.551**	0.118	0.485
	Isometric knee flexion	0.497	**0.504**	−0.177	0.254
**Neuromuscular control**	Static single-leg balance	−0.208	−0.122	**0.892**	−0.106
	Landing error scoring system	−0.319	−0.127	**0.772**	0.107
**Fatigue index**	Knee flexion fatigue index	−0.147	−0.047	−0.079	**0.854**
	Knee extension fatigue index	0.217	0.245	0.509	**0.614**
Eigenvalue		4.247	3.989	2.084	1.625
Variance (%)		30.335	28.492	14.884	11.604
Cumulative variance (%)		30.335	58.827	73.710	85.315
*KMO* = 0.318, Bartlett’s *x* = 253.318 (*p* < 0.001).					

**Table 3 ijerph-18-08053-t003:** The effect of the strongest factor of the Lower-Extremity Assessment Protocol on the International Knee Documentation Committee score.

**Dependent Variable**	***B***	***S.E.***	***β***	***t***	***F***	***p***	***VIF***	***R*** **^2^**	***_adj_*** ***R*** **^2^**	***R*****^2^** ***Change***
(Constant)	43.85	9.52		4.60						
Functional task ^a^	0.03	0.11	0.66	3.14	9.88 *	0.008	1.00	0.43	0.39	0.43

* *p* < 0.01. ^a^ Strength, neuromuscular control, and fatigue factors were excluded from stepwise regression.

**Table 4 ijerph-18-08053-t004:** The effect of the strongest component of the Lower-Extremity Assessment Protocol on the International Knee Documentation Committee score.

Dependent Variable	*B*	*S.E.*	*β*	*t*	*F*	*p*	*VIF*	*R* ^2^	*_adj_R* ^2^	*R* ^2^ *Change*
(Constant)	42.82	8.78		4.88						
Single hop ^a^	0.26	0.08	0.70	3.55	12.57 *	0.004	1.00	0.49	0.45	0.49

* *p* < 0.01. ^a^ The triple hop, cross-over hop, and 6 m timed hop were excluded from stepwise regression.

## Data Availability

Not applicable.

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
