# Peer review of "Determination of the Strongest Factor and Component in a Relationship between Lower-Extremity Assessment Protocol and Patient-Oriented Outcomes in Individuals with Anterior Cruciate Ligament Reconstruction: A Pilot Study"

_ijerph, 2021, doi:10.3390/ijerph18158053_

Round 1
Reviewer 1 Report
The authors have revised their manuscript and improved the quality of the presentation of their study. The critical point of group differences due to differing graft choices is now clearly stated in the manuscript.
Author Response
Dear Editor 1.
We sincerely appreciate your review and comments for our manuscript in all reviews round.
Reviewer 2 Report
Thank you for your kind revision of your work. I could see the improvement. The quality of current version is fairly okay.
Author Response
Dear Editor 2.
We sincerely appreciate your review and comments for our manuscript in all reviews round.
This manuscript is a resubmission of an earlier submission. The following is a list of the peer review reports and author responses from that submission.
Round 1
Reviewer 1 Report
The authors present an interesting study aiming at a validation of the components of the rather new lower extremity assessment protocol (LEAP) for RTP after ACL reconstruction and trying to simplify the LEAP by determining the most important component within the most influential factor. It remains unclear, if the participants in the current study actually used this protocol for RTP or if participants returned to sports within the one to five years follow-up after ACLR and therefor, before participating in this study. Moreover, it is not clearly stated that this was primary ACLR or if further surgeries to the knees of the included participants had been performed before.
Since the authors use the standardized and well established IKDC the research of Villa et al could be included in introduction and/or discussion, which assessed factors influencing subjective IKDC scores during recovery from ACLR and found them to be significantly higher in younger patients. Villa FD, Ricci M, Perdisa F, Filardo G, Gamberini J, Caminati D, Villa SD. Anterior cruciate ligament reconstruction and rehabilitation: predictors of functional outcome. Joints. 2016 Jan 31;3(4):179-85. doi: 10.11138/jts/2015.3.4.179 . PMID: 26904523 ; PMCID: PMC4739537.
Regarding RTP/RTS a wide variety of tests have been proposed, which might indicate that no clear consensus exists regarding these test batteries. Therefor, this research and especially factor analyses might be crucial to develop an easy and fast standardized test battery, without a large quantity of performed tests being partly redundant.
Prior literature has stated that the interpretation of these test results might be complicated, as the rehabilitation progress may be influenced by graft selection as well as the patients’ age, sex, and physical activity level (Csapo R, Pointner H, Hoser C, Gföller P, Raschner C, Fink C. Physical Fitness after Anterior Cruciate Ligament Reconstruction: Influence of Graft, Age, and Sex. Sports (Basel). 2020 Mar 6;8(3):30. doi: 10.3390/sports8030030 . PMID: 32155933 ; PMCID: PMC7183074.). Graft choice has not yet been reported for the current study, but seems to be crucial. Larger knee extensor strength deficits were found in isokinetic strength tests when reconstruction was performed with QT-autograft, whereas knee flexor strength was more strongly affected when a hamstring grafts was used. (Csapo R, Pointner H, Hoser C, Gföller P, Raschner C, Fink C. Physical Fitness after Anterior Cruciate Ligament Reconstruction: Influence of Graft, Age, and Sex. Sports (Basel). 2020 Mar 6;8(3):30. doi: 10.3390/sports8030030 . PMID: 32155933 ; PMCID: PMC7183074.).
In summary, it can be said that the Conclusio of the current study must at least be questioned relating to the points mentioned and therefore needs further thorough revision.
Author Response
Thank you very much for considering our research paper.
Please see the attachment.

Reviewer 2 Report
Dear Authors,
Thanks for giving me the chance to read this manuscript, “The most Influential Factors and Components of Lower-Extremity Assessment Protocol to Determine Return to Play for Patients after Anterior Cruciate Ligament Reconstruction: a pilot study”. Regarding the Lower-Extremity Assessment Protocol (LEAP) assesses multi-dimensional aspects of a patient with anterior cruciate ligament (ACL) injury, it is needed to reduce dimensionality to effectively assess patients which would definitely help to understand the return to play for patients. This study was to analyze the behavior of 15 patients. The result showed the 14 components of LEAP were categorized into four factors (functional task, muscle strength, neuromuscular control, and fatigue) that accounted for 83.8% of the cumulative variance by factor analysis.
Generally, it is a timely and interesting topic, and the paper is well written and designed with a clear direction toward research questions. However, there are some points worth noting which must be appropriately addressed.
- Sample size (Major)
As a quantitative study that aims to conduct factor analysis, a total of 15 individuals was definitely not adequate.
According to the prior research (Mundfrom et al., 2005), they recommended that, “suggested minimums for sample size include from 3 to 20 times the number of variables and absolute ranges from 100 to over 1,000.”
Thus, the current sample size is too far from adequacy. The authors need to justify the reason why 15 participants are enough for conducting a factor analysis from both theoretical and mathematical perspectives OR increase sample size instead.
Ref:
Mundfrom, D. J., Shaw, D. G., & Ke, T. L. (2005). Minimum sample size recommendations for conducting factor analyses. International Journal of Testing, 5(2), 159-168.
- Method
EFA was conducted in this study. In order to confirm the current dimensions found in this study, a confirmative factor analysis (CFA) via another sample was needed to verify this finding.
To sum up, I personally like this paper and its contributions. However, the sampling and method-related issues are needed to be introduced for a rigorous Hope these suggestions help.
Author Response

(The authors gave the same response as above.)

Reviewer 3 Report
This is an interesting study, in general, well written. I found some major design problems that must be improved and some minor issues across tables and methods that need attention. Bellow the authors will find a list of these.
Major:
The manuscript indicates the aim of validating LEAP, although there is no repeatability/validity analysis being performed. This aim must be reformulated.
The evaluator used motion-analysis Kinovea software for scoring. There are no details of the angle analysis or the scoring mentioned. This is a key part of the analysis and it can not be missed. Also Kinovea is a 2D analysis, which has limitations, therefore, care must be taken when videotaping (standardising the distance to the subject for example), was any of this done?, and care interpreting these results (should be mentioned in limitations).
The aim at the end of introduction is to 'simplify LEAP, this doesnt appear in the abstract, is this an important aim that need to be mentioned? or should be just removed?
Minor:
Why is this a pilot study in the title? it was not mentioned again in the paper.
IKDC questionnaire is mentioned in abstract but not explained in introduction, please do.
It is mentioned in the abstract 'to reduce dimensionality' please explain further
there are 2 exclusion criteria, is it 'and' or 'or' these two criteria?
Table 1: mention the units on the title.
Figure 1: describe the 3 parts of photo D. This figure is not mentioned in the text.
Tables 2 onwards: all abbreviations and units must be explained in full.
Rephrase the paragraph starting 'According to the results of current research...' it doesnt read well.
Give a citation for quadriceps test and patient satisfaction in Discussion section.
Author Response

(The authors gave the same response as above.)

Round 2
Reviewer 1 Report
The authors have partially revised their manuscript and provided effort to improve it all in all. Still some comments have not been adequaetly adressed. E.g. in the exclusion criteria the statement " a history of surgery in lower extremities except for ACLR" still doesn´t clearly exclude revision surgery which should definitely be excluded due to clear group differences.
Simply stating that only autografts were used, doesn´t clear up the already known differences eg. between hamstring and quadriceps tendon with regard to the mentioned rtp tests. Relevant group differences have not been clearly stated.
Reviewer 2 Report
The authors have addressed my quesitons. I am happy to recommend this paper.